# An Artificial Neural Network Approach to Predict the Effects of Formulation and Process Variables on Prednisone Release from a Multipartite System

**DOI:** 10.3390/pharmaceutics11030109

**Published:** 2019-03-07

**Authors:** Arthur Manda, Roderick B. Walker, Sandile M. M. Khamanga

**Affiliations:** Division of Pharmaceutics, Faculty of Pharmacy, Rhodes University, Grahamstown 6140, South Africa; arthur.artlin@gmail.com (A.M.); R.B.Walker@ru.ac.za (R.B.W.)

**Keywords:** prednisone, Response Surface Methodology, artificial neural networks, multi-layer perceptron, multiple-unit, extrusion-spheronization, USP Apparatus 2, Box–Behnken, reversed-phase high-performance liquid chromatography

## Abstract

The impact of formulation and process variables on the in-vitro release of prednisone from a multiple-unit pellet system was investigated. Box-Behnken Response Surface Methodology (RSM) was used to generate multivariate experiments. The extrusion-spheronization method was used to produce pellets and dissolution studies were performed using United States Pharmacopoeia (USP) Apparatus 2 as described in USP XXIV. Analysis of dissolution test samples was performed using a reversed-phase high-performance liquid chromatography (RP-HPLC) method. Four formulation and process variables viz., microcrystalline cellulose concentration, sodium starch glycolate concentration, spheronization time and extrusion speed were investigated and drug release, aspect ratio and yield were monitored for the trained artificial neural networks (ANN). To achieve accurate prediction, data generated from experimentation were used to train a multi-layer perceptron (MLP) using back propagation (BP) and the Broyden-Fletcher-Goldfarb-Shanno (BFGS) 57 training algorithm until a satisfactory value of root mean square error (RMSE) was observed. The study revealed that the in-vitro release profile of prednisone was significantly impacted by microcrystalline cellulose concentration and sodium starch glycolate concentration. Increasing microcrystalline cellulose concentration retarded dissolution rate whereas increasing sodium starch glycolate concentration improved dissolution rate. Spheronization time and extrusion speed had minimal impact on prednisone release but had a significant impact on extrudate and pellet quality. This work demonstrated that RSM can be successfully used concurrently with ANN for dosage form manufacture to permit the exploration of experimental regions that are omitted when using RSM alone.

## 1. Introduction

Prednisone is effective for treating certain types of arthritis, severe allergic reactions, multiple sclerosis, lupus and pathologies that affect the lungs, skin, stomach and intestines [1]. In addition, prednisone is prescribed for serious pathologies such as cancer and as an adjunct to pneumonia therapy in patients with acquired immunodeficiency syndrome [2,3]. Regardless of all therapeutic benefits, the oral delivery of prednisone is associated with many challenges. Prednisone is a borderline Biopharmaceutics Classification System class I compound. Therefore, poor solubility and low dissolution rate in gastrointestinal fluids often result in low bioavailability [1]. The low tolerance level which depends on dose strength results in fluctuations in overall absorption [4]. Furthermore, prednisone has harsh effects on the gastrointestinal tract (GIT) [5], in addition to an extremely bitter taste which needs to be masked to achieve better palatability [6].

Multi-particulate dosage forms offer significant advantages over conventional technologies in many aspects, particularly as they exhibit a lower incidence of gastrointestinal irritation due to decreased local concentration of the active pharmaceutical ingredient (API) in the GIT following oral administration [7]. Furthermore, lower individual variability in plasma concentrations is observed when compared to tablets since there is a reduced risk of dose dumping [7]. In addition, the presence of many individual units increases the surface area leading to improved solubility and bioavailability [8]. The use of discrete units also offers a simple solution to minimizing potential API–excipient interactions and the free-flowing nature of pellets facilitates reproducible capsule filling, content uniformity and dosing [9]. Moreover, loading the dosage form into gelatin capsules is an easy way to mask the bitter taste of an API.

Extrusion-spheronization was used to manufacture prednisone pellets. The technique is relatively simple and has been used to prepare pellets containing a variety of compounds using different excipient combinations [10,11]. Extrusion-spheronization is a multi-step process that involves several formulation parameters and process settings that may affect the properties of pellets [12], therefore it is critical to use a RSM approach in order to save time and minimize costs.

RSM is a set of mathematical and statistical techniques designed to fit experimental data to empirical models [13]. RSM uses Design of Experiments to diversify all significant parameters, simultaneously, and then combine the results through use of a mathematical model. The resulting model is used to analyze, predict and optimize the significant variables [14]. Optimization by means of RSM is divided into six stages: (i) selection of independent variables and output responses; (ii) selection of an experimental design; (iii) execution of experiments according to the selected experimental matrix; (iv) fitting of experimental data to mathematical and statistical models; (v) verification of the model and production of response surface plots; and (vi) identification of the optimal conditions for the system [14]. The use of RSM is effective in generating information in a relatively short time using a minimum number of experiments. ANN are computational algorithms that are designed to simulate the human brain when analyzing data by learning from experience [15]. Similar to the human brain, ANN are capable of processing multi-dimensional, non-linear, clustered and imprecise information. Thus, ANN can be used to decode complicated real world problems that are sometimes challenging to evaluate using statistical approaches [16]. ANN models can be classified based on their functions viz., associating networks, feature extracting networks and non-adaptive networks [17]. Associating networks are suitable for predicting dissolution kinetics and other dosage form parameters. Following training, associating networks are capable of mapping the relationship between formulation, process parameters and output responses such as dissolution rate. These networks can then be used to predict the rate of dissolution of new formulations prepared with completely different excipient concentrations and process settings. ANN may also be used to establish the optimized formulation in which case the trained neural network is used to select optimum formulation composition and process parameters that produce the best results [17]. 

In this study, ANN were used to complement RSM. Generally, RSM requires the specification of a polynomial function such as a linear, first-order interaction or second-order quadratic for regression. In addition, the number of terms in the polynomial function is limited to the number of experimental design points and the selection of an appropriate polynomial equation can be tedious since each response requires an individual polynomial equation. The ANN approach is quite flexible and can model complex relationships without the need for complicated equations, and is capable of exploring regions that are otherwise omitted when using statistical approaches. With ANN, all data are used, potentially making ANN-based models more accurate [13]. An integrated RSM and ANN approach compliments the deficits observed when RSM is used alone.

The objective of this work was to evaluate the impact of microcrystalline cellulose concentration, sodium starch glycolate concentration, spheronization time and extrusion speed on the in-vitro release of prednisone from a multiple-unit pellet system. In-vitro dissolution testing was used to assess prednisone release since it is an integral part of pharmaceutical development and is often a key component of routine quality control testing [18]. In addition, the study evaluated the effectiveness of an integrated RSM and ANN approach to formulation development and testing.

## 2. Materials and Methods

### 2.1. Materials

Prednisone powder was purchased from Skyrun (Taizhou, China). Microcrystalline Cellulose (Comprecel^®^ M102 D+) was purchased from Mingtai chemicals (Taoyuan Hsien, Taiwan). Eudragit^®^ RL 30 D was donated by Rohm Pharma (Darmstadt, Germany). Sodium starch glycolate, talc, magnesium stearate, Tween^®^ 80 were donated by Aspen Pharmacare (Port Elizabeth, South Africa). Polyethylene Glycol (PEG) 400 was purchased from Merck chemicals (Johannesburg, South Africa). Hydrochlorothiazide was purchased from Skyrun (Taizhou, China). HPLC far UV grade acetonitrile (ROMIL-SpS^™^) was purchased from Microsep (Port Elizabeth, South Africa). The HPLC grade water used for analyses was prepared using a Milli-Q^®^ Academic A10 water purification system (Millipore^®^, Bedford, MA, USA), consisting of an Ionex^®^ ion-exchange cartridge and a quantum EX-ultrapore organex cartridge, which was fitted with a 0.22 μm Millipak^®^ 40 sterile filter (Millipore^®^, Bedford, MA, USA).

### 2.2. Preparation of Pellets

Prednisone pellets were manufactured using extrusion-spheronization. Design Expert^®^ software (Version 8.0.7.1, State-Ease Inc., Minneapolis, MN, USA) was used to generate multivariate experiments for a Box–Behnken design and the formulation and process parameters used are listed in Table 1. The target batch size was 50 g. For all batches the concentration of prednisone was 4% *w*/*w* and the concentration of talc and magnesium stearate was 1.5% and 0.5% *w*/*w* respectively. Screening studies revealed that formulation variables such as microcrystalline cellulose concentration (*X*_1_) and sodium starch glycolate concentration (*X*_2_) and process variables such as spheronization time (*X*_3_) and extrusion speed (*X*_4_) were the main factors that impacted prednisone release. These factors were varied to investigate their impact on pellet production and performance. The remaining excipients viz., Tween^®^ 80 (T), PEG 400 (P) and Eudragit^®^ RL 30 D (E) were used in the same ratio, but actual quantities varied depending on the microcrystalline cellulose concentration and sodium starch glycolate concentration used. All excipients used for manufacture are generally safe and have been used in other dosage forms intended for oral administration [19,20,21].

Prednisone powder, microcrystalline cellulose (Comprecel^®^ M102 D+), sodium starch glycolate, talc and magnesium stearate were separately weighed on a Mettler AG 135 top loading balance (Mettler Instruments, Zurich, Switzerland) according to working formulas for each batch (Table 1). The powders were transferred to a Kenwood Multi-Pro FP580 planetary mixer (Kenwood Ltd. Maraisburg, South Africa) and blended for 4 min. Tween^®^ 80, PEG 400 and a 50% *v*/*v* aqueous dilution of Eudragit^®^ RL 30 D were added to the planetary mixer and the concentrations blended until a uniform paste had formed. Water was then added slowly with blending until a powder mass of optimal wetness had formed. To ensure uniform mixing, materials were repeatedly scraped from the walls of the mixing bowl during the granulation process. The resultant mass was then passed through a Model 20 Caleva^®^ extruder (Schlueter, Neustadt am Ruebenberge, Germany) fitted with co-rotating impellers and a screen of aperture pore size 1 mm (diameter). Extrusion was conducted at speeds 25, 30 or 35 rpm. Sufficient time was allowed to harvest the maximum yield of extrudate that were then transferred to a Caleva^®^ MBS 250 spheronizer (Schlueter, Neustadt am Ruebenberge, Germany) immediately following extrusion and spheronized for 1, 2 or 3 min. The spheronizer was fitted with a 250 mm diameter crosshatched friction plate of 3 × 3 mm pitch and 1.2 mm depth. Preliminary experiments revealed that high speeds resulted in a low pellet yield due to formation of fine material and therefore a low speed of 642 rpm was used throughout this study.

Following manufacture, the pellets were collected and dried at 40 °C for 6 h in a size one hotbox oven (Gallenkamp^®^, Weiss Technik, Loughborough, UK). The pellets were collected and stored in tightly sealed 100 mL glass containers (Lasec^®^ Solutions, Cape Town, South Africa) and some were filled into opaque yellow size 1 gelatin capsules for further investigation. Each capsule was loaded with pellets equivalent to 5 mg prednisone and stored in a cool, dry place away from light. The aspect ratio (*Y*_1_), yield (*Y*_2_), prednisone release at 15 min (*Y*_3_), 30 min (*Y*_4_), 45 min (*Y*_5_) and 60 min (*Y*_6_) were monitored as output responses.

### 2.3. Dissolution Test Conditions

In-vitro release studies were performed using the method described in USP XXIV [22] using a Model SR 8 PLUS USP Apparatus 2 (Hanson Research, Chartsworth, CA, USA) that was fitted with an Autoplus™ Multifill™, maximizer syringe fraction collector and a digitally controlled water circulation heater.

Accurately weighed pellets (125 mg) containing 5 mg of prednisone were loaded into size 1 gelatin capsules. To prevent capsules from floating, spiral capsule sinkers (Hanson Research, Chartsworth, CA, USA) were used. The capsules were dropped into 500 mL of HPLC grade water maintained at 37 ± 0.5 °C. The paddles were set to rotate at 50 rpm and at 5, 10, 15, 20, 25, 30, 45, and 60 min aliquots (5 mL) of dissolution fluid were automatically collected and placed into test tubes for further analysis. After each collection equal volumes of dissolution media were replaced into the vessels to maintain sink conditions. Aliquots (1 mL) of each sample were analyzed using a validated RP-HPLC method developed our laboratory.

### 2.4. HPLC Analysis

Analysis of dissolution samples was undertaken using a Waters^®^ Model e2695 Alliance Separations Module (Waters^®^, Milford, MA, USA), equipped with an auto-sampler, degasser, solvent delivery module and a Model 2489 UV/Vis (Waters^®^, Milford, MA, USA) detector set at 254 nm. The stationary phase was a Phenomenex Synergi^™^ Polar-RP 80Å 250 mm × 4.6 mm i.d. × 4 µm column (Separations^®^, Randburg, South Africa) maintained at a temperature of 25 °C. The mobile phase consisted of water and acetonitrile in a 65%:35% *v*/*v* ratio. For all analyses a 1 mL aliquot was added to 0.5 mL of internal standard solution, hydrochlorothiazide (100 µg/mL), mixed and analyzed. 20 µL samples were injected and monitored at a constant flow rate of 1 mL/min and wavelength of 254 nm. Data processing was achieved using Waters Empower^®^ 3 software (Waters^®^, Milford, MA, USA). The RP-HPLC method was validated according to ICH guidelines [23] and was found to be linear, precise, accurate and specific. A coefficient of correlation (*R*^2^) of 0.9985 was observed for linearity studies over the concentration range 1–100 µg/mL. Intra-day (repeatability) and inter-day (intermediate) precision studies produced relative standard deviations ≤0.89% and ≤1.79% respectively. Prednisone recovery during accuracy studies was approximately 100% with a relative standard deviation ≤0.264% and percent bias ≤0.04%. Method specificity was determined by analyzing commercially available Be-Tab^®^ Prednisone tablets and the recovery of prednisone was approximately 100% for all samples.

### 2.5. Scanning Electron Microscopy (SEM)

A scanning electron microscope (VEGA LMU^©^, Tescan, Brno, Czechoslovakia Republic) was used to observe the shape and surface morphology of the pellets. Samples were mounted onto aluminum stubs using double sided adhesive tape and sputter coated with gold for 20 min in a Hitachi vacuum coating unit. Coated samples were viewed at an accelerated voltage of 20 kV and a probe current of 20 nA at 960 × 1280 pixels. The aspect ratio was used to assess pellet sphericity. The aspect ratio is defined as the ratio of maximum Feret diameter to the diameter perpendicular to the maximum Feret diameter [24]. Ideally a value of 1 indicates a perfect sphere, but any ratio ≤1.2 is considered acceptable [24]. The images generated using SEM were transferred to analySIS docu^®^ software (Olympus, Hamburg, Germany) for size analysis and the mean aspect ratio of pellets (*n* = 6) determined. The aspect ratio was calculated using Equation (1):(1)$$Aspect\;ratio=\frac{d_{max}}{d_{90}}$$
where, *d_max_* = maximum Feret diameter and *d*_90_ = Feret diameter perpendicular to *d_max_*.

### 2.6. Yield

The yield was calculated using Equation (2) and was presented as a percent of the mass of pellets manufactured relative to the theoretical yield [25].
(2)$$Yield\;\left(\%\right)=\frac{Actual\;yield\;\left(g\right)}{Theoretical\;yield\;\left(g\right)}\times 100$$

### 2.7. Response Surface Modelling

Box and Behnken proposed a three level design for fitting response surfaces [26]. The design is a combination of a two factorial and incomplete block design and the result is a cube shaped, revolving design that consists of a central point and points at the middle of each edge [26,27]. Box–Behnken designs are either rotatable or nearly rotatable and are very efficient in terms of the number of experimental runs required to elucidate a solution [27]. In addition, Box–Behnken designs do not contain combinations for which all factors are simultaneously at their highest or lowest levels hence these designs are useful in avoiding experiments performed under extreme conditions for which unsatisfactory outcomes may result [28]. A Box–Behnken design was used to generate multivariate experiments for pellet manufacture as it is rotatable and more efficient. The purpose of the study was to find an optimum experimental region therefore a design that provided an equal precision of estimation in all directions was preferred [29].

A four-factor Box–Behnken design was used for the optimization procedure. This design is suitable for exploring quadratic response surfaces and constructing second order polynomial models. Each independent variable was coded at three levels viz., −1, 0 and +1. A total of twenty nine experimental runs were conducted (Equation (3)):(3)$$N=2k\;\left(k-1\right)+C_{0}$$
where, *N* = the number of experiments, *k* = the number of factors and $C_{0}$ = the number of central points. 

The non-linear quadratic model generated by the design is of the form presented as Equation (4):(4)$$Y=\beta_{0}+\beta_{i}X_{1}+\beta_{i}X_{2}+\beta_{i}X_{3}+\beta_{i}X_{4}+\beta_{ij}X_{1}X_{2}+\beta_{ij}X_{1}X_{3}+\beta_{ij}X_{1}X_{4}+\beta_{ij}X_{2}X_{3}\;+\beta_{ij}X_{2}X_{4}+\beta_{ij}X_{3}X_{4}+\beta_{ii}{X_{1}}^{2}+\beta_{ii}{X_{2}}^{2}+\beta_{ii}{X_{3}}^{2}+\beta_{ii}{X_{4}}^{2}$$
where, *y* = the measured response, $\beta_{0}$ = an intercept, $\beta_{i}$ = a coefficient of a first order term, $\beta_{ij}$ = a coefficient of a second order interaction and $\beta_{ii}$ = a coefficient of a quadratic interaction [30]. 

### 2.8. Artificial Neural Network

RSM is widely used to examine and optimize input variables for experiment design and model development by mapping a response surface over a particular region of interest and identifying the optimum conditions required to achieve target specifications. ANN may be used to complement RSM when studying complex systems [31,32,33]. One major drawback of RSM based methods is that they limit the number of experiments by eliminating runs in undesirable experimental domains. In order to bridge the gap, we used an ANN based model since it is potentially more accurate as all experimental data are included [34]. ANN learn from experimentally generated data, formally referred to as the training or learning set then validated using test data in a recall phase [35].

The fundamental unit of an ANN is the neuron [35]. Neurons in each layer are connected to those in the next layer through weighted connections known as synapses [35,36]. The topology of the neural network was designated as 4-h-6 representing four input neurons for the independent variables microcrystalline cellulose concentration (*X*_1_), sodium starch glycolate concentration (*X*_2_), spheronization time (*X*_3_) and extrusion speed (*X*_4_); h representing the number of neurons in one hidden layer and six representing the output responses aspect ratio (*Y*_1_), yield (*Y*_2_) and prednisone released at 15 min (*Y*_3_), 30 min (*Y*_4_), 45 min (*Y*_5_) and 60 min (*Y*_6_).

The data were entered into Statistica Neural Network software (Stat Soft, Inc., Tulsa, OK, USA) using BP and a BFGS 57 training algorithm. Both MLP and radial basis function (RBF) were investigated, however the MLP consistently produced R^2^ values closer to 1.0 with lower mean absolute error (MAE) values. The number of neurons in the hidden layer was altered between 2 and 15 and training was conducted by a trial and error search approach until a minimum RMSE was observed. RMSE, MAE and R^2^ were used to establish the efficiency of the neural network and are defined in Equations (5–7) [37]: (5)$$R^{2}=\frac{\sum_{i=1}^{n}(Y_{i,p}-Y_{i,e})}{\sum_{i=1}^{n}{\left(Y_{i,p}-Y_{e}\right)}^{2}}$$
(6)$$MAE=\frac{i}{n}\sum_{i=1}^{n}\left(Y_{i,e}-Y_{i,p}\right)$$
(7)$$RMSE=\sqrt{\frac{\sum_{i=1}^{n}{\left(Y_{i,e}-Y_{i,p}\right)}^{2}}{n}}$$
where, *n* = the total number of experiments, *Y_i,e_* = the experimental value of the *i*th experiment, *Y_i,p_* = the predicted value of the *i*th experiment by the model and *Y_e_* = the mean of experimentally determined values.

## 3. Results and Discussion

### 3.1. RSM Modeling

The second-order polynomial equations in terms of coded factors are reported in Equations (8–13):*Y*_1_ = 1.25 − 0.052*X*_1_ + 5.833e^−003^*X*_2_ − 0.066*X*_3_ + 1.667e^−003^*X*_4_ − 0.095*X*_1_*X*_2_ − 2.5e^−003^*X*_1_*X*_3_ + 0.11*X*_1_*X*_4_ + 0.03*X*_2_*X*_3_ − 7.5e^−003^*X*_2_*X*_4_ + 0.020*X*_3_*X*_4_ + 0.095*X*_1_^2^ − 0.094*X*_2_^2^ − 0.029*X*_3_^2^ − 0.040*X*_4_^2^(8)
*Y*_2_ = 61.84 + 7.87*X*_1_ − 0.4*X*_2_ + 2.42*X*_3_ − 0.71*X*_4_ + 3.53*X*_1_*X*_2_ − 8.90*X*_1_*X*_3_ + 12.55*X*_1_*X*_4_ + 0.82*X*_2_*X*_3_ − 0.65*X*_2_*X*_4_ + 8.43*X*_3_*X*_4_ − 4.35*X*_1_^2^ − 11.27*X*_2_^2^ + 6.56*X*_3_^2^ + 3.07*X*_4_^2^(9)
*Y*_3_ = 61.76 − 20.11*X*_1_ + 15.85*X*_2_ − 4.33*X*_3_ − 1.42*X*_4_ + 1.32*X*_1_*X*_2_ + 4.65*X*_1_*X*_3_ − 5.85*X*_1_*X*_4_ + 2.05*X*_2_*X*_3_ − 6.38*X*_2_*X*_4_ − 7.68*X*_3_*X*_4_ − 11.35*X*_1_^2^ − 2.73*X*_2_^2^ − 3.17*X*_3_^2^ − 2.61*X*_4_^2^(10)
*Y*_4_ = 75.00 − 16.40*X*_1_ + 12.73*X*_2_ − 4.33*X*_3_ − 1.91*X*_4_ + 8.45*X*_1_*X*_2_ + 1.80*X*_1_*X*_3_ − 3.50*X*_1_*X*_4_ + 1.05*X*_2_*X*_3_ − 5.90*X*_2_*X*_4_ − 5.98*X*_3_*X*_4_ − 11.35*X*_1_^2^ + 0.096*X*_2_^2^ − 2.67*X*_3_^2^ − 1.02*X*_4_^2^(11)
*Y*_5_ = 80.84 − 12.43*X*_1_ + 10.70*X*_2_ − 4.33*X*_3_ − 1.47*X*_4_ + 8.55*X*_1_*X*_2_ + 0.4*X*_1_*X*_3_ − 4.57*X*_1_*X*_4_ − 2.55*X*_2_*X*_3_ − 8.55*X*_2_*X*_4_ − 4.17*X*_3_*X*_4_ − 9.92*X*_1_^2^ + 4.54*X*_2_^2^ − 3.57*X*_3_^2^ − 0.037*X*_4_^2^(12)
*Y*_6_ = 84.54 − 7.17*X*_1_ +7.15*X*_2_ − 3.67*X*_3_ − 1.80*X*_4_ + 10.68*X*_1_*X*_2_ + 0.17*X*_1_*X*_3_ − 4.10*X*_1_*X*_4_ − 1.50*X*_2_*X*_3_ − 5.82*X*_2_*X*_4_ − 2.42*X*_3_*X*_4_ − 11.60*X*_1_^2^ + 6.13*X*_2_^2^ − 3.52*X*_3_^2^ + 1.65*X*_4_^2^(13)

The analysis of variance (ANOVA) data generated from studies is listed in Table 2. The *R*^2^, adjusted (Adj) *R*^2^ and coefficient of variation (CV) values are important parameters to consider when explaining model adequacy and fitness [38,39,40,41,42]. Results from this study revealed that most *R*^2^ values were close to 1 indicating good model fitness. The low CV and standard deviation (SD) values indicated better accuracy and precision of the model. All ratios for adequate (Adeq) precision were ≥5.53 which indicates an adequate signal. The model for aspect ratio (*p* > 0.05) was deemed insignificant. All other models exhibited *p* < 0.05 and were significant. F-values for the model varied with each response under investigation and showed the usefulness of the model. 

The results from these studies revealed that microcrystalline cellulose concentration and sodium starch glycolate concentration were the most significant factors impacting prednisone release. An inverse relationship was observed between microcrystalline cellulose concentration and sodium starch glycolate concentration. The rate of dissolution was more rapid when the concentration of sodium starch glycolate was at 2% *w*/*w* and the concentration of microcrystalline cellulose was at 50% *w*/*w*. The 3D response surface plot depicted in Figure 1 provides a visual representation of the impact of microcrystalline cellulose concentration and sodium starch glycolate concentration on prednisone release at 30 min. For all other time points investigated the same trend and conformation was observed.

Microcrystalline cellulose was incorporated into the formulation as a diluent and pelletization aid. Microcrystalline cellulose is highly hygroscopic [43], and has a tendency to adsorb and retain large volumes of water during granulation imparting adequate plasticity to the wet mass thereby facilitating pelletization [43]. Increasing the concentration of microcrystalline cellulose retarded the rate of prednisone release. A possible explanation for this phenomenon may be postulated using the crystallite-gel model proposed by Kleinebudde [44] that suggests that microcrystalline cellulose particles disaggregate eventually forming single crystallites due to shear stress forces encountered during granulation and extrusion. In the presence of a liquid, these crystallites form a crystallite-gel held together in a framework by cross linked hydrogen bonds at the amorphous ends of the molecules. Hydrogen bonding is even more apparent when water is used as a granulating fluid and an even stronger matrix is formed. Since microcrystalline cellulose forms the gel, the strength and extent of adhesive forces that maintain pellet integrity are highly dependent on the fraction of microcrystalline cellulose used in a formulation. In addition, the water required to form the gel increases with increasing microcrystalline cellulose concentration further influencing the extent of hydrogen bonding. When water eventually evaporates during drying, new hydrogen bonds are formed, and the interweaving of microcrystalline cellulose fibers continues to hold the pellet structure together. Due to this phenomenon the time taken for pellets to dissociate increases with increasing microcrystalline cellulose concentration due to extensive bonding resulting in slower prednisone release.

Sodium starch glycolate is a super-disintegrant [45] and was incorporated into the formulation to facilitate pellet dissociation. Sodium starch glycolate exerts its effect via a swelling mechanism [45] when exposed to aqueous media. The swelling effect overcomes inter-particulate adhesive forces that hold the pellet together resulting in dissociation into smaller fragments thereby releasing prednisone trapped in each pellet [46]. Changes in sodium starch glycolate concentration drastically impacted pellet disintegration dynamics. Increasing the amount of sodium starch glycolate used improved the rate and extent of prednisone released and vice versa. Visual observation of pellets during dissolution testing revealed that the magnitude of swelling was greatest and disintegration time slowest for pellets containing higher sodium starch glycolate concentration. The smaller fragments have a large surface area to volume ratio which favors dissolution, therefore changes in sodium starch glycolate concentration impacted the rate and extent of fragmentation proportionally, ultimately affecting the dissolution rate of prednisone. These results are consistent with the findings of Alves-Silva et al. [11] and Kilor et al. [47] who observed an improvement in dissolution rate of immediate release benznidazole and aceclofenac pellets respectively when sodium starch glycolate was incorporated.

Pharmacopoeial guidelines were used to determine the suitability of prednisone release profiles. According to pharmacopoeial guidelines, immediate release oral dosage forms must release at least 80% of the active ingredient within 30 min [18]. Previous studies have shown that incorporating surfactants alone or in combination with glycerides can improve the aqueous solubility of hydrophobic drugs [48,49,50]. Tween^®^ 80 was added to the formulation to facilitate solubilization of prednisone. PEG 400 was used as a pore-forming agent to ensure the ingress of aqueous fluid into pellets via capillary action [48]. In addition, PEG 400 was incorporated to improve the solubility of prednisone as it is hydrophilic nature [48]. A 50% *v*/*v* aqueous dilution of Eudragit^®^ RL 30 D was prepared to reduce the viscosity and permit easy spraying during the granulation process and was incorporated into the formulation to improve solubility and to impart tensile strength. Talc and magnesium stearate were added as anti-adherents to minimize friction during manufacturing. This combination of excipients resulted in rapid wetting and hydration of the pellets, exposing the disintegrant to aqueous media within seconds. As a result, disintegration was rapid and dissolution of prednisone from most formulations tested complied to the pharmacopoeial limits.

Spheronization time and extrusion speed had minimal impact on prednisone release but had a significant impact on extrudate and pellet quality. Generally, an ideal extrudate must be non-adhesive to itself and must be rigid enough to retain the shape imposed by a die, yet be sufficiently brittle to be broken into short lengths by the spheronizer without disintegrating completely [44]. An extrusion speed of 25 rpm was found to produce extrudate conforming to these criteria. The quality of the resultant extrudate was not only a function of extrusion speed, but rather a combination of the formulation composition and ideal speed to confer sufficient shear to produce extrudates that met the aforementioned criteria. Formulations containing high Tween^®^ 80 and PEG 400 concentration produced extrudates that were visually appealing however, the high viscosity resulting from Tween^®^ 80 and PEG 400 inclusion resulted in sticking, prompting a degree of coalescence and globulation during spheronization especially when longer residence times were used. The impact of spheronization time on pellet quality varied, however, sufficient sphericity was observed after the first minute. Formulation composition had an impact on pellet size. A spheronization time of 1 min was sufficient to produce pellets and aspect ratios ≤1.20 were calculated suggesting acceptable pellet sphericity.

Whilst the yield and aspect ratio were not the main attributes investigated in the study they were monitored to determine the efficiency of production and if acceptable sphericity of pellets was achieved. As depicted in the 3D response surface plot (Figure 2), the yield increased with increasing microcrystalline cellulose concentration since it is a pelletization aid. The yield also improved with increasing spheronization time possibly due to more extrudate converting to a pellet state. The 3D response surface plot for aspect ratio is depicted in Figure 3. The model for aspect ratio was not significant, however a visual representation of the impact of some model terms on aspect ratio is also presented. 

### 3.2. ANN Modelling

The optimum neural network was established as a MLP 4-6-6 network with four input neurons representing input variables, microcrystalline cellulose concentration (*X*_1_), sodium starch glycolate concentration (*X*_2_), spheronization time (*X*_3_) and extrusion speed (*X*_4_), six neurons in a single hidden layer used for computation and six output neurons representing output responses, aspect ratio (*Y*_1_), yield (*Y*_2_), prednisone release at 15 min (*Y*_3_), 30 min (*Y*_4_), 45 min (*Y*_5_) and 60 min (*Y*_6_). The ANN was identified using BP and a BFGS 57 training algorithm. A schematic representation of the optimum MLP neural network identified is depicted in Figure 4. Statistica neural network software automatically partitioned experimental data into a training, test and cross-validation set to avoid over training.

The optimum number of neurons in the hidden layer was identified following a systematic trial and error method. The number of neurons in the hidden layer was altered and each time weights resulting in the least RMSE between experimental and predicted values were selected until a satisfactory RMSE was obtained. The responses predicted by the optimum ANN are listed with experimental responses in Table 3.

The results reveal that data predicted by the ANN model were in close agreement with experimentally obtained data, indicating high prediction accuracy of the ANN model. The closeness of predicted and experimental results from runs 4, 17 and 22 suggest that they may have been used for cross validation. The Box–Behnken design used for this study used five replicate center points. As a result, replicate ANN predictions were observed for runs 9, 10, 13, 15 and 29. A comparison of the predictive capacity of RSM and ANN for all output responses is summarized in Table 4.

The results reveal that the ANN model for aspect ratio and prednisone release exhibited better prediction accuracy than the RSM model as indicated by higher *R*^2^ values and lower MAE and RMSE values observed. The higher predictive accuracy of the ANN is attributed to its ability to process multi-dimensional, non-linear and clustered information whereas RSM is limited to use of a second order polynomial [39]. The generation of an optimum ANN is a multi-step calculation process, that is repeated until a desirable error is achieved whereas a response surface model is based on a single step calculation. One major limitation of the Statistica neural network software is that it only contains a single hidden layer. Even though our results revealed better modelling accuracy with ANN they can potentially be improved with an ANN that contains more hidden layers for better computation.

The optimized dosage form was expected to comply with pharmacopoeial specifications for immediate release solid oral dosage forms ie. ≥80% prednisone to be released within 30 min of commencing dissolution testing [18]. In addition, the target aspect ratio was ≤1.20. Pellets with aspect ratios within the specified range exhibit good flow properties and have a large surface area to volume ratio which facilitates reproducible capsule filling and improves solubility. Following experimentation and data modelling, the results were analyzed to assess desirability and a list of excipient concentrations for the optimized formulation that resulted in optimum prednisone release with an acceptable aspect ratio and yield is summarized in Table 5. The extrusion speed was 25 rpm and the spheronization time 2 min. The resultant formulation may be suitable for treating acute conditions.

During screening studies, prednisone concentrations between 4–10% *w*/*w* were assessed. The results obtained when prednisone concentration was at 4% *w*/*w* were comparable to when the concentration was at 10% *w*/*w*. The dissolution profile of the optimized formulation is depicted in Figure 5. Spherical (A) and porous (B) pellets were produced as depicted in Figure 6. Formulation 12 (Figure 6) is an example of failed pelletization since elongated, non-uniform, irregular and dumb-bell shaped pellets (C) were formed possibly due to the short spheronization time or insufficient shear stress or inappropriate excipient concentrations.

## 4. Conclusions

An immediate release multiple-unit pellet system for prednisone was successfully developed and manufactured using the extrusion-spheronization method. The availability of an alternative oral drug delivery system for prednisone will address some of the challenges associated with oral delivery of prednisone observed when conventional technologies are used. From the parameters investigated, only microcrystalline cellulose concentration and sodium starch glycolate concentration were found to have a significant impact on prednisone release. An inverse relationship was observed for microcrystalline cellulose concentration and sodium starch glycolate concentration. Increasing the microcrystalline cellulose concentration resulted in a retarded rate of prednisone release and vice versa, whereas increasing sodium starch glycolate concentration improved the rate of prednisone release and vice versa. All other excipients and process settings contributed to the production of an immediate release formulation that performed within pharmacopoeial specifications for such dosage forms. 

The dissolution profile for the optimized formulation revealed gradual release of prednisone from the multipartite system suggesting that use in-vivo may result in lower incidences of gastrointestinal irritation due to the decreased local concentration of prednisone in the GIT following oral administration. In addition, the gradual release of prednisone addresses the challenge of dose dumping that is observed with conventional technologies. The presence of many individual units in a multipartite system results in an increased surface area leading to improved solubility and dissolution as shown by complete prednisone release within 60 min of commencing dissolution testing. Moreover, the prednisone pellets were loaded into gelatin capsules to produce the final dosage form with the benefit of masking the bitter taste of prednisone. 

This work illustrates that RSM and ANN can be successfully used concurrently during dosage manufacture. A Box–Behnken design was used to generate multivariate experiments that established the relationship between input variables and output responses. The knowledge gained from Box–Behnken design experiments was used to train a neural network until an optimum ANN architecture was obtained. The optimum ANN was used to predict the effects of formulation and process variables on prednisone release and can be used for any future predictions. The study also proved that ANN has better modelling accuracy than RSM.

## Figures and Tables

**Figure 1 pharmaceutics-11-00109-f001:**
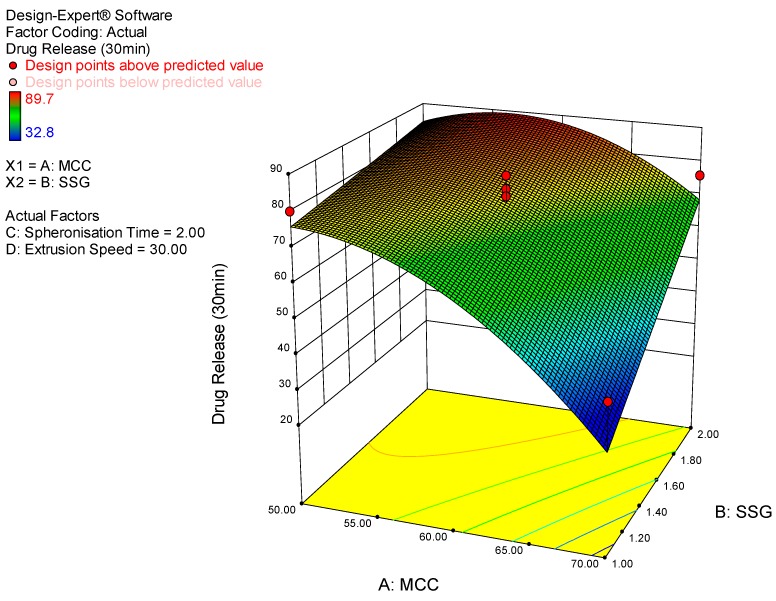
3D response surface plot depicting the impact of microcrystalline cellulose and sodium starch glycolate concentration on prednisone release at 30 min.

**Figure 2 pharmaceutics-11-00109-f002:**
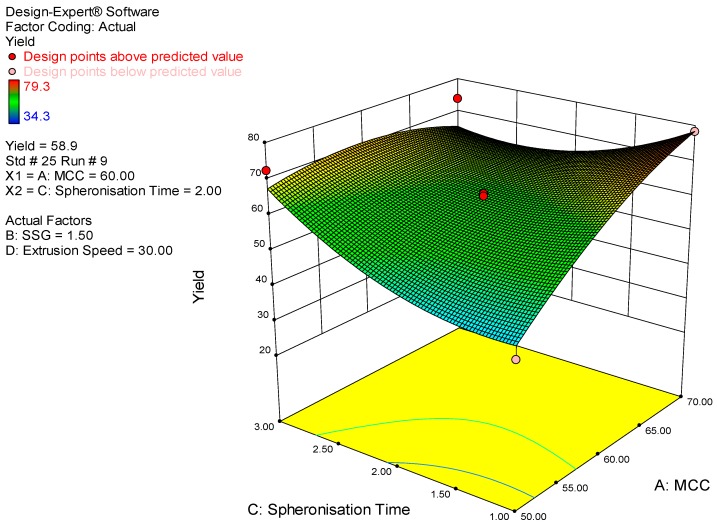
3D response surface plot depicting the impact of microcrystalline cellulose concentration and spheronization time on % yield.

**Figure 3 pharmaceutics-11-00109-f003:**
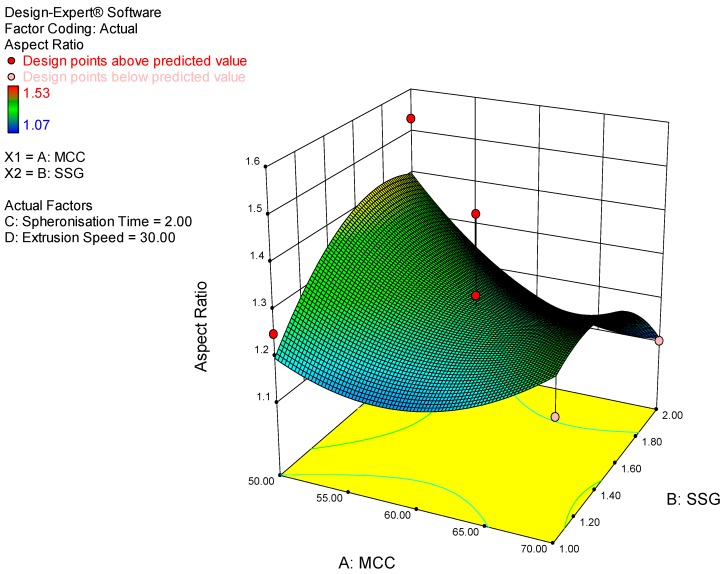
3D response surface plot depicting the impact of microcrystalline cellulose and sodium starch glycolate concentration on aspect ratio.

**Figure 4 pharmaceutics-11-00109-f004:**
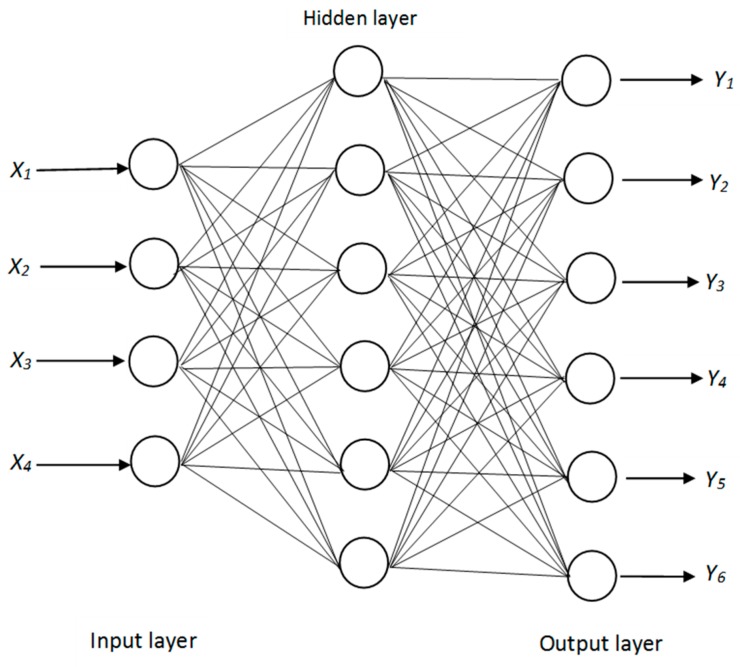
Schematic representation of the optimum 4-6-6 multi-layer perceptron artificial neural network.

**Figure 5 pharmaceutics-11-00109-f005:**
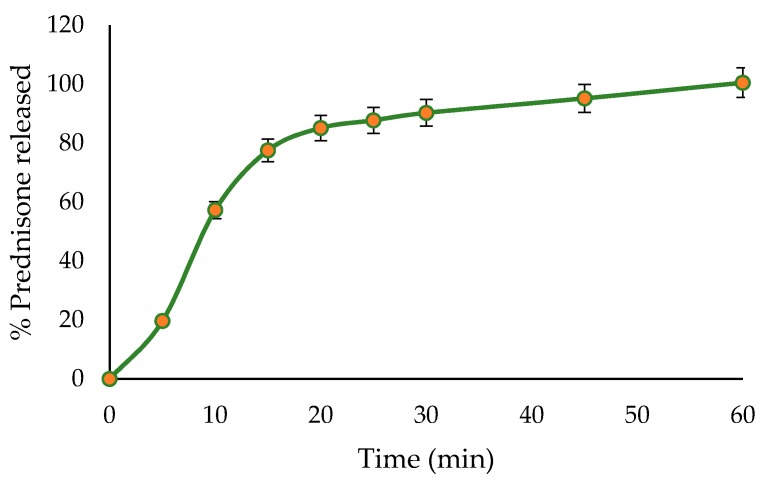
In-vitro release profile from optimized prednisone pellets (*n* = 3).

**Figure 6 pharmaceutics-11-00109-f006:**
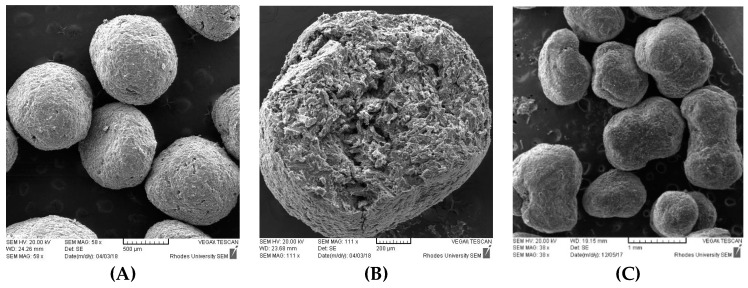
SEM images for the (**A**) optimized formulation; (**B**) cross section of the optimized formulation revealing a porous internal structure; and (**C**) elongated, irregular, non-uniform and dumb-bell shaped pellets for formulation 12.

**Table 1 pharmaceutics-11-00109-t001:** Formulation and process variables used to manufacture prednisone pellets.

Run	Microcrystalline Cellulose (*X*_1_) % *w*/*w*	Sodium Starch Glycolate (*X*_2_) % *w*/*w*	Spheronization Time (*X*_3_) min	Extrusion Speed (*X*_4_) rpm	Tween^®^ 80 (T) % *w*/*w*	PEG 400 (P) % *w*/*w*	Eudragit^®^ RL 30 D (E) % *w*/*w*
1	60	2.0	1	30	12.8	6.4	12.8
2	60	1.0	2	25	13.2	6.6	13.2
3	60	1.0	1	30	13.2	6.6	13.2
4	70	1.0	2	30	9.20	4.6	9.20
5	70	1.5	2	25	9.00	4.5	9.00
6	50	1.5	1	30	17.0	8.5	17.0
7	60	2.0	3	30	12.8	6.4	12.8
8	60	2.0	2	25	12.8	6.4	12.8
9	60	1.5	2	30	13.0	6.5	13.0
10	60	1.5	2	30	13.0	6.5	13.0
11	70	1.5	3	30	9.00	4.5	9.00
12	50	2.0	2	30	16.8	8.4	16.8
13	60	1.5	2	30	13.0	6.5	13.0
14	60	1.5	1	35	13.0	6.5	13.0
15	60	1.5	2	30	13.0	6.5	13.0
16	60	1.5	3	35	13.0	6.5	13.0
17	60	2.0	2	35	12.8	6.4	12.8
18	60	1.0	3	30	13.2	6.6	13.2
19	50	1.0	2	30	17.2	8.6	17.2
20	50	1.5	2	25	17.0	8.5	17.0
21	50	1.5	2	35	17.0	8.5	17.0
22	50	1.5	3	30	17.0	8.5	17.0
23	70	1.5	2	35	9.00	4.5	9.00
24	70	1.5	1	30	9.00	4.5	9.00
25	60	1.5	3	25	13.0	6.5	13.0
26	70	2.0	2	30	8.80	4.4	8.80
27	60	1.5	1	25	13.0	6.5	13.0
28	60	1.0	2	35	13.2	6.6	13.2
29	60	1.5	2	30	13.0	6.5	13.0

**Table 2 pharmaceutics-11-00109-t002:** ANOVA data generated for RSM model.

Response	*R* ^2^	Adj *R*^2^	Pred *R*^2^	Adeq Precision	CV %	*F*-Value	*p*-Value	SD
*Y* _1_	0.6594	0.3188	−0.4673	5.530	9.12	1.94	0.1144	0.11
*Y* _2_	0.9128	0.8256	0.5137	13.129	8.42	10.47	0.0001	5.00
*Y* _3_	0.8885	0.7769	0.4885	10.775	17.33	7.97	0.0002	9.28
*Y* _4_	0.8606	0.7213	0.3201	10.612	13.03	6.18	0.0008	8.97
*Y* _5_	0.8405	0.6811	0.1800	10.297	10.90	5.27	0.0018	8.41
*Y* _6_	0.8143	0.6286	−0.0071	9.671	9.35	4.39	0.0046	7.62

**Table 3 pharmaceutics-11-00109-t003:** Experimental and ANN-predicted responses.

Run	Aspect Ratio		Yield %		Drug Release % (15 min)		Drug Release % (30 min)		Drug Release % (45 min)		Drug Release % (60 min)	
	*Y_1EXP_*	*Y_1ANN_*	*Y_2EXP_*	*Y_2ANN_*	*Y_3EXP_*	*Y_3ANN_*	*Y_4EXP_*	*Y_4ANN_*	*Y_5EXP_*	*Y_5ANN_*	*Y_6EXP_*	*Y_6ANN_*
1	1.09	1.10	55.7	45.3	75.3	76.3	88.2	87.6	100.1	95.0	100.3	98.6
2	1.07	1.22	54.8	62.2	31.1	32.1	53.3	56.0	66.8	68.8	84.6	78.0
3	1.21	1.19	58.3	64.7	37.1	34.2	56.0	57.8	65.0	66.7	75.2	78.4
4	1.20	1.21	47.5	47.7	24.6	24.6	39.8	39.8	51.3	51.3	59.4	59.4
5	1.19	1.18	58.4	62.5	32.8	25.4	51.8	46.0	66.5	63.5	74.6	74.0
6	1.40	1.43	41.2	39.4	80.4	78.2	89.7	82.8	90.7	91.3	90.2	89.4
7	1.08	1.07	56.4	57.8	71.5	67.9	82.8	84.6	86.2	94.0	90.5	96.5
8	1.09	1.07	52.5	50.0	77.5	80.3	88.2	89.5	95.1	95.1	99.1	99.3
9	1.16	1.25	58.9	57.7	63.4	63.3	78.8	77.5	85.2	84.6	87.9	87.9
10	1.17	1.25	63.2	57.7	47.2	63.3	62.3	77.5	71.9	84.6	77.6	87.9
11	1.19	1.23	73.9	65.7	18.5	24.6	32.8	40.0	43.3	51.6	51.1	60.4
12	1.53	1.50	34.3	36.8	80.2	80.4	82.0	83.8	91.3	91.4	80.9	79.0
13	1.45	1.25	60.8	57.7	64.0	63.3	76.7	77.5	82.8	84.6	84.8	87.9
14	1.27	1.26	60.4	64.0	68.7	65.2	83.5	77.4	86.2	79.3	86.7	86.0
15	1.27	1.25	62.8	57.7	72.6	63.3	82.6	77.5	86.6	84.6	86.5	87.9
16	1.24	1.07	79.3	77.0	50.8	45.4	67.1	79.9	73.9	88.0	80.8	94.9
17	1.11	1.18	56.0	55.8	64.1	65.4	80.0	79.9	85.3	86.7	90.5	88.7
18	1.08	1.08	55.7	61.3	25.1	25.9	46.4	46.0	61.4	61.9	71.7	75.4
19	1.25	1.28	39.2	40.1	63.7	76.3	79.9	80.1	85.3	85.3	84.0	85.3
20	1.53	1.52	70.9	78.3	58.9	80.4	77.1	88.5	80.9	80.7	81.5	83.9
21	1.17	1.12	36.4	38.9	67.3	63.0	72.3	73.7	77.1	78.5	77.2	76.6
22	1.15	1.07	72.4	71.2	55.6	57.5	72.1	73.0	76.6	76.1	75.4	76.1
23	1.28	1.23	74.1	66.5	17.8	24.6	33.0	40.1	44.4	51.6	53.9	60.7
24	1.45	1.27	78.3	67.5	24.7	25.3	43.2	44.2	55.8	55.1	65.2	66.8
25	1.09	1.16	62.1	59.1	70.7	73.3	82.3	82.5	85.4	92.4	87.0	94.2
26	1.10	1.17	56.7	62.6	46.4	55.6	75.7	77.7	91.5	88.8	99.0	92.1
27	1.20	1.09	76.9	56.2	57.9	75.8	74.8	87.2	81.0	94.9	83.2	98.5
28	1.12	1.07	60.9	54.5	43.2	43.6	68.7	67.0	91.2	89.9	99.3	95.8
29	1.18	1.25	63.5	57.7	61.6	63.3	74.6	77.5	77.7	84.6	85.9	87.9

*Y_EXP_*—experimental value; *Y_ANN_*—predicted by ANN model.

**Table 4 pharmaceutics-11-00109-t004:** Comparison of RSM and ANN.

Parameter	Aspect Ratio		Yield		Drug Release % (15 min)		Drug Release % (30 min)		Drug Release % (45 min)		Drug Release % (60 min)	
	*Y_1RSM_*	*Y_1ANN_*	*Y_2RSM_*	*Y_2ANN_*	*Y_3RSM_*	*Y_3ANN_*	*Y_4RSM_*	*Y_4ANN_*	*Y_5RSM_*	*Y_5ANN_*	*Y_6RSM_*	*Y_6ANN_*
*R* ^2^	0.659	0.797	0.913	0.849	0.889	0.934	0.861	0.946	0.841	0.934	0.814	0.906
MAE	0.012	0.007	24.97	41.51	86.13	55.94	80.46	33.29	70.72	32.42	58.10	31.90
RMSE	0.110	0.081	4.997	6.443	9.281	7.479	8.970	5.770	8.410	5.694	7.622	5.648

**Table 5 pharmaceutics-11-00109-t005:** Optimum formulation composition for prednisone pellets.

Material	Concentration %	Function
Prednisone	4	Active ingredient
Tween 80	12.8	Surfactant, Solubilizer.
Polyethylene glycol 400	6.4	Pore-former, solubilizer, Imparts hydrophilicity.
Eudragit^®^ RL 30 D (50 % aqueous dilution)	12.8	Improves pellet tensile strength, Imparts hydrophilicity.
Comprecel^®^ M102 D+	60	Bulking agent, spheronization aid.
Sodium starch glycolate	2	Disintegrant.
Talc	1.5	Glidant, anti-adherent.
Magnesium stearate	0.5	Lubricant, anti-adherent.

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
