# Peer review of "An Artificial Neural Network Approach to Predict the Effects of Formulation and Process Variables on Prednisone Release from a Multipartite System"

_pharmaceutics, 2019, doi:10.3390/pharmaceutics11030109_

Reviewer 1 Report

This study describes the development of an immediate release multiple-unit pellet system of prednisone produced by extrusion-spheronization using a novel integrate approach of RSM and ANN. The article shows a good example of the application of this development strategy. Some formatting flaws need to be corrected so it may be suitable for publication.

Abstracts

Methodological details can be suppressed, whereas more results from the study should be described such as how the excipients concentrations affect the drug release.  

Introduction

-p1, L27-37: Please, remove the abbreviations used just once throughout the text such as AIDS, and MUPS.  

-p1, L35-37: This sentence is not appropriate to the first paragraph of the introduction. Please, remove the sentence “In this study…”

- p2, L48-51: Please, provide more update references and avoid references from dissertations such as the reference 18. A recommend the use of this reference: Alves-Silva, Ihatanderson, et al. "Preparation of benznidazole pellets for immediate drug delivery using the extrusion spheronization technique." Drug development and industrial pharmacy 43.5 (2017): 762-769. In addition, use this reference to support sodium starch glycolate effect on drug delivery of pellets (p11, L173-175).

Methods

-p3, L87-89: References are missing.

-p3, L104: Please, revise your manuscript in order to eliminate typing errors and excessive spaces. For example, see p6, L18: Theoretical…

-p3, L117: Drug dose placed in each cube are missing.

-p4, L139: Please, described shortly some of the validation results obtained such as linearity correlation, specificity data, and precision coefficients of variation.   

-p6, L1: What was the number of images used to assess the pellet sphericity?

-p6, L20-26: Provide a better justification of the choice for the Box-Behnken design. For this use the reference: de Alencar, Rodrigo Gomes, et al. "Compacted multiparticulate systems for colon-specific delivery of ketoprofen." AAPS PharmSciTech 18.6 (2017): 2260-2268.

-p7, L69: Make the information more precise: “microcrystalline cellulose concentration”; “sodium starch glycolate concentration”. Make the correction throughout the manuscript

-p7, L47: Considering information from Table 1, the Table 2 is unnecessary. 

-p7, L48-88: Make it short. 

Results and discussion

-p9, L91-97: Methodological explanation must be removed from this section.

-p9, L110-130: Make it short. Basic statistical explanations are unnecessary.

-p9, L132-135: Remove these sentences.

 -p13, L217-226: Please, remove methodological information from this section.

-p16, L10-11: Expand the discussion about the construction of an optimal formulation condition. To do this, use the reference Pinho, Ludmila, et al. "Dissolution Enhancement in Cocoa Extract, Combining Hydrophilic Polymers through Hot-Melt Extrusion." Pharmaceutics 10.3 (2018): 135.

Author Response

RESPONSE TO REVIEWER 1’S COMMENTS

Point 1: This study describes the development of an immediate release multiple-unit pellet system of prednisone produced by extrusion-spheronization using a novel integrate approach of RSM and ANN. The article shows a good example of the application of this development strategy. Some formatting flaws need to be corrected so it may be suitable for publication.

Response 1: The manuscript has been reformatted.

Point 2: Abstracts

Methodological details can be suppressed, whereas more results from the study should be described such as how the excipients concentrations affect the drug release.  

Response 2: Methodological details have been reduced and more results have been presented, especially the effects of excipients concentration on drug release.

INTRODUCTION

Point 3: -p1, L27-37: Please, remove the abbreviations used just once throughout the text such as AIDS, and MUPS.  

Response 3: Abbreviations (AIDS and MUPS) have been removed.

Point 4: -p1, L35-37: This sentence is not appropriate to the first paragraph of the introduction. Please, remove the sentence “In this study…”

Response 4: The sentence beginning with, “In this study …” has been removed.

Point 5: - p2, L48-51: Please, provide more update references and avoid references from dissertations such as the reference 18. A recommend the use of this reference: Alves-Silva, Ihatanderson, et al. "Preparation of benznidazole pellets for immediate drug delivery using the extrusion spheronization technique." Drug development and industrial pharmacy 43.5 (2017): 762-769. In addition, use this reference to support sodium starch glycolate effect on drug delivery of pellets (p11, L173-175).

Response 5: Reference 18 has been removed. Recent references included. Recommended reference (Alves-Silva Ihatanderson et al.) included.

METHODS

Point 6: -p3, L87-89: References are missing.

Response 6: Screening studies were conducted to elucidate significant factors.

Point 7: -p3, L104: Please, revise your manuscript in order to eliminate typing errors and excessive spaces. For example, see p6, L18: Theoretical…

Response 7: The manuscript has been revised and all errors corrected.

Point 8: -p3, L117: Drug dose placed in each cube are missing.

Response 8: Dosage has been included. [Accurately weighed pellets (125 mg) containing 5 mg of prednisone were loaded into gelatin capsules…]

Point 9: -p4, L139: Please, described shortly some of the validation results obtained such as linearity correlation, specificity data, and precision coefficients of variation.   

Response 9: Validation results have been described.

Point 10: -p6, L1: What was the number of images used to assess the pellet sphericity?

Response 10: Six.

Point 11: -p6, L20-26: Provide a better justification of the choice for the Box-Behnken design. For this use the reference: de Alencar, Rodrigo Gomes, et al. "Compacted multiparticulate systems for colon-specific delivery of ketoprofen." AAPS PharmSciTech 18.6 (2017): 2260-2268.

Response 11: Another reason for using Box-Behnken design provided. It is also efficient compared to other designs. 

Point 12: -p7, L69: Make the information more precise: “microcrystalline cellulose concentration”; “sodium starch glycolate concentration”. Make the correction throughout the manuscript

Response 12: The recommended format has been used throughout the manuscript.

Point 13: -p7, L47: Considering information from Table 1, the Table 2 is unnecessary. 

Response 13: Table 2 has been removed.

Point 14: -p7, L48-88: Make it short. 

Response 14: The information on ANN has been presented in a shorter and more precise form.

RESULTS AND DISCUSSION

Point 15: -p9, L91-97: Methodological explanation must be removed from this section.

Response 15: Methodological details have been removed.

Point 16: -p9, L110-130: Make it short. Basic statistical explanations are unnecessary.

Response 16: Statistical explanations have been removed.

Point 17: -p9, L132-135: Remove these sentences.

Response 17: The sentences have been removed.

Point 18: -p13, L217-226: Please, remove methodological information from this section.

Response 18: Methodological details have been removed.

Point 19: -p16, L10-11: Expand the discussion about the construction of an optimal formulation condition. To do this, use the reference Pinho, Ludmila, et al. "Dissolution Enhancement in Cocoa Extract, Combining Hydrophilic Polymers through Hot-Melt Extrusion." Pharmaceutics 10.3 (2018): 135.

Response 19: A more articulate discussion has been provided.

Reviewer 2 Report

The manuscript by Manda et al explored artificial neural network (ANN) approach to predict the effects of formulation and process variables on the prednisone release. The authors found that microcrystalline cellulose and sodium starch glycolate content influence the release profile of prednisone in vitro. This manuscript could considered further for publications only after the following comments are addressed.

RSM is mentioned for the first time in line 68, but there is no prior explanation nor extension of the abbreviation used. Moreover, the justification of using an integrated RSM and ANN, instead of using either RSM or ANN, should be described.

Prenidsone is for in vivo use (line 27-30) but the formulation described in this manuscript consists of a variety of components (e.g. Eudragit, Tween 80, PEG400, etc) with questionable biocompatibility/toxicity. The authors should add explaination whether they are safe for therapeutic use.

What are the basis of determining the "optimum" formulation as in Table 6?

In Table 6, Prednisone 4% is rather low. How to increase the prednisone content to, perhaps, 10-20%?

It looks like that the data obtained in Figure 5 was from a single experiment. Duplicate or even triplicate is often conducted to obtain sufficiently confident data.

In Figure 6, the authors mentioned that the third image contains dumbbell shaped pellets based only on less than 10 pellets. I expect that the image is zoomed out to see more pellets otherwise I would think of "Irregular shape" as more appropriate in this regards.

The effects of microcrystalline cellulose and sodium starch glycolate content should be described in the Conclusion. If those two variables produce good effect, why aren't their composition increased in the formulation?

Author Response

RESPONSE TO REVIEWER 2’S COMMENTS

Point 1: The manuscript by Manda et al explored artificial neural network (ANN) approach to predict the effects of formulation and process variables on the prednisone release. The authors found that microcrystalline cellulose and sodium starch glycolate content influence the release profile of prednisone in vitro. This manuscript could considered further for publications only after the following comments are addressed.

Response 1: All comments have been addressed.

Point 2: RSM is mentioned for the first time in line 68, but there is no prior explanation nor extension of the abbreviation used. Moreover, the justification of using an integrated RSM and ANN, instead of using either RSM or ANN, should be described.

Response 2: More information on RSM has been included in the abstract and introduction. Reasons for using both RSM and ANN have also been articulated.

Point 3: Prednisone is for in vivo use (line 27-30) but the formulation described in this manuscript consists of a variety of components (e.g. Eudragit, Tween 80, PEG400, etc) with questionable biocompatibility/toxicity. The authors should add explanation whether they are safe for therapeutic use.

Response 3: Small amounts of these excipients were incorporated which is safe. These excipients have been used previously for different oral dosage forms and we have provided a few relevant references for such use.

Point 4: What are the basis of determining the "optimum" formulation as in Table 6?

Response 4: The United States Pharmacopeia lists recommended performance criteria for immediate release dosage forms. Therefore, our designated optimum formulation had to meet the stipulated release criteria. In addition aspect ratio had to be ≤ 1.20 and acceptable yield.

Point 5: In Table 6, Prednisone 4% is rather low. How to increase the prednisone content to, perhaps, 10-20%?

Response 5: (i) It is more cost effective to have a low concentration of the API, (ii) the target dose was 5 mg per capsule and using a 4 % w/w concentration produced an amount of pellets which could fit well in size 1 gelatin capsules, (iii) Screening studies were conducted prior to optimization studies and incorporation of higher concentrations of prednisone produced results which were comparable to formulations containing 4 % w/w prednisone and (iv) use of 4 % w/w produced the desired release profile within the USP recommended time.

Point 6: It looks like that the data obtained in Figure 5 was from a single experiment. Duplicate or even triplicate is often conducted to obtain sufficiently confident data.

Response 6: Triplicate experiments were conducted and information revealing such has now been added.

Point 7: In Figure 6, the authors mentioned that the third image contains dumbbell shaped pellets based only on less than 10 pellets. I expect that the image is zoomed out to see more pellets otherwise I would think of "Irregular shape" as more appropriate in this regards.

Response 7: The sentence has been restructured to include other shapes too eg: irregular, dumb-bell, elongated …

Point 8: The effects of microcrystalline cellulose and sodium starch glycolate content should be described in the Conclusion. If those two variables produce good effect, why aren't their composition increased in the formulation?

Response 8: The effects microcrystalline cellulose and sodium starch glycolate concentration have been described in the conclusion. The intended release criteria was achieved using the listed optimized concentrations therefore we did not see any reason to alter formulation composition.

Round  2

Reviewer 1 Report

Satisfactory corrections were provided.